# Intracellular dynamics of the Sigma-1 receptor observed with super-resolution imaging microscopy

Sergei Kopanchuk[1], Edijs Vavers[2,3]*, Santa Veiksina[1], Kadri Ligi[1], Liga Zvejniece[2], Maija Dambrova[2,3], Ago Rinken[1]*

**1** University of Tartu, Institute of Chemistry, Tartu, Estonia, **2** Latvian Institute of Organic Synthesis, Riga, Latvia, **3** Riga Stradins University, Riga, Latvia

* ago.rinken@ut.ee (AR); edijs.vavers@farm.osi.lv (EV)

## Abstract

Sigma-1 receptor (Sig1R) is an endoplasmic reticulum (ER)-related membrane protein, that forms heteromers with other cellular proteins. As the mechanism of action of this chaperone protein remains unclear, the aim of the present study was to detect and analyze the intracellular dynamics of Sig1R in live cells using super-resolution imaging microscopy. For that, the Sig1R-yellow fluorescent protein conjugate (Sig1R-YFP) together with fluorescent markers of cell organelles were transfected into human ovarian adenocarcinoma (SK-OV-3) cells with BacMam technology. Sig1R-YFP was found to be located mainly in the nuclear envelope and in both tubular and vesicular structures of the ER but was not detected in the plasma membrane, even after activation of Sig1R with agonists. The super-resolution radial fluctuations approach (SRRF) performed with a highly inclined and laminated optical sheet (HILO) fluorescence microscope indicated substantial overlap of Sig1R-YFP spots with KDEL-mRFP, slight overlap with pmKate2-mito and no overlap with the markers of endosomes, peroxisomes, lysosomes, or caveolae. Activation of Sig1R with (+)-pentazocine caused a time-dependent decrease in the overlap between Sig1R-YFP and KDEL-mRFP, indicating that the activation of Sig1R decreases its colocalization with the marker of vesicular ER and does not cause comprehensive translocations of Sig1R in cells.

## Introduction

The non-opioid sigma-1 receptor (Sig1R) has been described as an intracellular protein modulator, a chaperone protein that localizes in the endoplasmic reticulum (ER) [1]. It was thought that, upon stimulation by agonists or cell stress, Sig1R can translocate from the ER to the plasma membrane, where it interacts with and affects the function of other receptors, ion channels, and kinases [2]. For example, atomic force microscopy imaging confirmed the direct binding of Sig1R to the hERG potassium channel within the plane of the plasma membrane [3]. However, APEX2-enhanced electron microscopy revealed that Sig1R localizes in the ER and nucleoplasmic reticulum but not in the plasma membrane [4]. To date, it is not fully understood how ER-resident Sig1R can translocate within the ER and interact with other

**Data Availability Statement:** All relevant data are within the manuscript and its Supporting Information files.

**Funding:** SV- Estonian Research Council grant (PSG230) EV- European Regional Development Fund Project No. 1.1.1.2/VIAA/2/18/376 "Sigma chaperone protein as a novel drug target" AR, EV - COST action CA 18133 ERNEST. The funders had no role in study design, data collection and analysis, decision to publish, or preparation of the manuscript.

**Competing interests:** The authors have declared that no competing interests exist.

proteins, especially those located on the plasma membrane. As described previously, it seems more likely that the interactions between Sig1R and plasma membrane-resident proteins could only be formed at the proximity between the ER and the plasma membrane [2]. The mechanism of action and intracellular dynamics of Sig1R remain unclear, hindering the understanding of the functional nature of Sig1R.

Several selective fluorescent Sig1R ligands have been developed for imaging in cells using microscopy techniques [5]. However, only a few studies have demonstrated the dynamics of Sig1R in live cells using a fluorescence imaging approach. For example, in live neuroblastoma-glioma NG108-15 cells, Sig1R dynamics have been demonstrated using Time-Lapse Fluorescence microscopy [6]. The movement of activated yellow fluorescent protein (YFP)-tagged Sig1R from lipid-enriched globules to tubular elements on the ER has been demonstrated with real-time monitoring by confocal microscopy [7]. The subcellular and functional dynamics of Sig1R have also been studied using live-cell imaging of fluorescently tagged wild-type and mutant Sig1R-YFP in mouse embryonic fibroblasts and Chinese hamster ovary cells [8]. Even so, the detection of Sig1R in cells is still technically challenging due to the lack of specific high-affinity Sig1R antibodies and fluorescently labeled probes [2]. Confocal microscopy with FRAP assays have provided the first hints about the agonist-dependent increased mobility of Sig1R-YFP [8]. However, the resolution limit of the assays does not allow us to obtain sufficiently detailed information about the protein-protein interactions or precise cellular mobility of Sig1R.

In recent years, several new methods in fluorescence microscopy have been developed, enabling the achievement of higher resolution than the diffraction limit of light. Most of these methods require sophisticated hardware and special experimental design, such as STED, SIM, and PALM [9], but there are also several algorithm-based methods, such as FAstLocalization algorithm based on a CONtinuous-space formulation (FALCON) [10] and the super-resolution radial fluctuations (SRRF) [11]. These later methods enable super-resolution with temporal resolution in the low-seconds range [10]. Recently, to achieve super-resolution level, protein retention expansion microscopy (pro-ExM) was applied to demonstrate Sig1R expression in mitochondria-associated membranes (MAMs) [12].

In the present study, we used SRRF to detect and analyze the intracellular dynamics of Sig1R in live cells after pharmacological activation with selective ligands and found time-dependent changes in the colocalization of Sig1R with KDEL-mRFP, a marker of the vesicular ER.

## Materials and methods

### Plasmid constructions and generation of BacMam viruses

BacMam technology was used for cotransfection of human ovarian adenocarcinoma (SK-OV-3) cells with Sig1R-YFP conjugate and fluorescent markers of the ER and mitochondria as we have described earlier [13]. The expression vector for pcDNA3.1(+)-Sig1R-YFP [14] was kindly provided by Prof. P. McCormick from the University of London. The Sig1R-YFP construct, under the control of the cytomegalovirus (CMV) promoter, was cloned into the pFastBac1 vector (Invitrogen Life Technologies) using restriction enzymes (Fermentas, Vilnius, Lithuania) RruI and NotI for pcDNA3.1(+) and NotI and Eco105I for pFastBac1, respectively. To ensure low promoter interference during BacMam virus amplification, the polyhedrin promoter was removed from the pFastBac1 vector.

The construct of a fluorescent marker targeting the ER lumen, KDEL-mRFP [15], was kindly provided by Prof. Erik L. Snapp from the Albert Einstein College of Medicine. The mitochondrial marker pmKate2-mito was obtained from Evrogen JSC (cat.# FP187).

The previous scheme was used for cloning of these markers into the pFastBac1 vector and production of recombinant bacmid DNA, with the exception that PciI and NotI restriction enzymes were used for pmRFP-N1 and for pmKate2-mito, while PciI and Eco105I were used for pFastBac1.

All generated pFastBac1 constructs were verified by sequencing. Then, the Bac-to-Bac baculovirus system was used according to the manufacturer's recommendations (Thermo Fisher Scientific, Invitrogen) to transform the obtained pFastBac constructs into DH10Bac competent cells for the production of recombinant bacmid DNAs. All obtained bacmid DNAs were PCR-verified and then transfected into Sf9 insect cells by using FuGENE 6 Transfection Reagent (Promega). BacMam viruses were harvested and further amplified to high-titer viral stocks (in the range of 108 virus particles per milliliter). Virus titers were determined by image-based cell-size estimation (ICSE) assay [16]. For long-term storage, viral stock aliquots were stored at -90˚C; working viral stocks were stored at 4˚C.

## SK-OV-3 cell culturing and transfection

SK-OV-3 cells (human ovarian adenocarcinoma, ATCC HTB-77) were maintained as a monolayer culture on cell culture treated 6-well multidishes (Thermo Scientific™, Bio-Lite) in McCoy's 5A Medium supplemented with 10% FBS, 2 mM L-glutamine, 100 U/ml penicillin, and 100 μg/ml streptomycin at 37˚C in humidified 5% $CO_2$ cell incubator.

For cell membrane preparations, SK-OV-3 cells were grown on 10 $cm^2$ treated Petri dishes (Thermo Scientific™, Nunc™). Recombinant BacMam baculoviruses of Sig1R-YFP were added to the cells (approximately 70% confluence) at an MOI of 3 and incubated for ≈24 h (medium was supplemented with 10 mM sodium butyrate (final concentration). An MOI = 3 was found to be optimal to achieve transfection of the majority of cells (Fig 1B) but not too high to initiate the generation of protein aggregates into cytoplasmic inclusion bodies.

For microscopy experiments, ≈24 h prior to imaging SK-OV-3 cells were seeded in 8-well glass-bottom imaging chambers (Zell Kontakt, Germany) at a density of 20 000–25 000 cells per well. Then the cells were transfected with appropriate BacMam baculoviruses (MOI ≈ 3), or FuGENE 6 transfection reagent was used for direct transfection. Plasmids of the markers of peroxisomes (mCherry-Peroxisomes-2), endosomes (mCherry-Endo-14), and lysosomes (mCherry-lysosomes-20) were a gift from Michael Davidson via Addgene (plasmids # 54520, # 55040 and # 55073, respectively). The marker of caveolae (Cav1-mRed) was a gift from Richard Pagano via Addgene (plasmid # 12681) [17]. Approximately 2 h prior to imaging, the culture medium was exchanged with photostable cell culture medium MEMO® with Supplement A (LiveLight™ products from Cell Guidance Systems).

## Immunofluorescence assays

Cell fixation and permeabilization for all experiments with antibodies were performed according to the earlier published protocol [18]. 3% v/v Glyoxal (Sigma-Aldrich, Cat#: 128465) in 20% ethanol (pH was adjusted to 4,5 with glacial acetic acid (Sigma-Aldrich)) was used for fixation. SK-OV-3 cells, grown in 8-well glass-bottomed imaging chambers and transfected with Sig1R-YFP BacMam as described above, were fixed with Glyoxal solution, keeping cells on ice for 30 min and then at room temperature for an additional 30 min. Glyoxal was neutralized with 100 mM $NH_4Cl$ (20 min), and then the cells were permeabilized and blocked with 0.1% Triton X-100 and 2.5% BSA in PBS (15 min). For labeling of the mitochondrial outer membrane, anti-Tom20 (D8T4N)—rabbit mAb (Cell Signaling Technologies, Cat#: 42406) was used and anti-PDI (RL90)—mouse mAb, (Abcam, Cat#: ab2792) was used as an ER marker. Cells were incubated 60 min with 1:200 dilutions of primary antibodies in permeabilization/

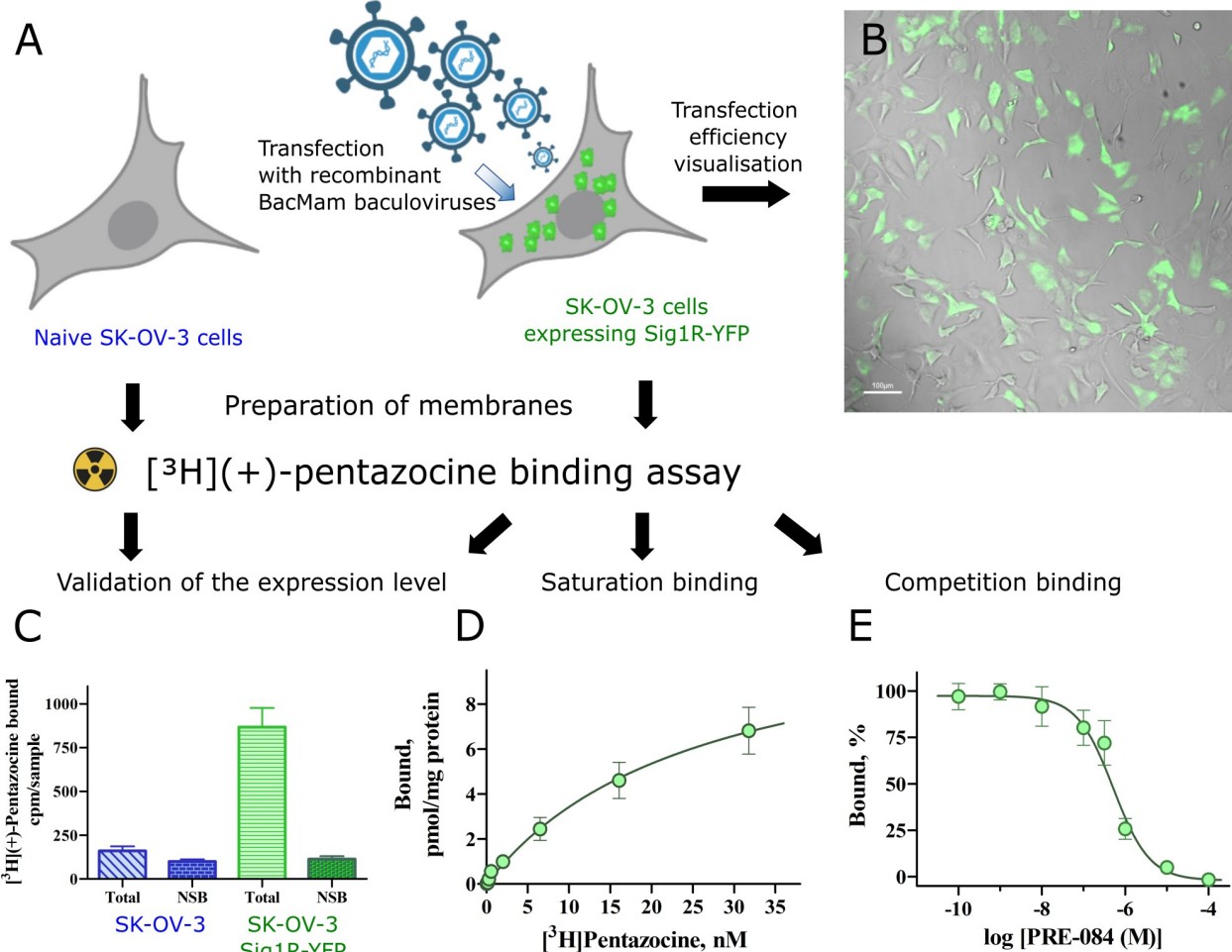

**Fig 1. Ligand binding properties of Sig1R-YFP in SK-OV-3 cells.** (A) Overview of binding experiments performed (created with Biorender.com). (B) SK-OV-3 cells were transfected with recombinant Bac-Mam baculoviruses expressing Sig1R-YFP (MOI = 3). The expression of the receptors was visualized with a 4× objective in brightfield (gray) and epifluorescence of YFP (green) channels. Scale bar = 100 μm. (C) Comparison of the binding of [3H](+)-pentazocine to membranes of SK-OV-3 and SK-OV-3-Sig1R-YFP cells. The data are presented as the means ± SEM (n = 6). NSB–nonspecific binding determined in the presence of 10 μM haloperidol. (D) Specific binding of [3H](+)-pentazocine to membranes of SK-OV-3-Sig1R-YFP cells. The specific binding (Bound) was defined as the difference between total and nonspecific binding, measured in the absence and presence of 10 μM haloperidol, respectively. The data are shown as the mean ± SEM from three independent experiments carried out in duplicate. (E) Competitive binding of [3H](+)-pentazocine and the selective Sig1R agonist PRE-084 in membranes of SK-OV-3-Sig1R-YFP cells. The data are shown as the mean ± SEM from four independent experiments carried out in duplicate. ***p < 0.001 in comparison with SK-OV-3 cells.

blocking solution and then washed three times for 5 min with the permeabilization/blocking solution. Then the cells were incubated for 60 min at room temperature with secondary antibodies (goat anti-rabbit (in case of anti-Tom20) or goat anti-mouse (in case of anti-PDI), both labeled abberior STAR RED (Abberior). Cells were washed 3 times for 5 min in PBS to remove unbound secondary antibodies and embedded in the Ibidi mounting medium (Ibidi, Cat#: 50001).

## Preparation of membranes from SK-OV-3 cells

The membranes from SK-OV-3 cells with and without Sig1R-YFP were prepared as described earlier [19] with some modifications. The cells were transferred from culture dishes into Dulbecco's PBS (2 mL per dish) and collected by centrifugation at 1000 ×g for 10 min at 4°C. The

cell pellet was homogenized with a tissue homogenizer (Coleparmer Labgen 125) for 30 s in ice-cold buffer containing 5 mM Tris, 5 mM $MgCl_2$, and Complete EDTA-free Protease Inhibitor Cocktail (Roche), pH = 7.4. The homogenate was then centrifuged at 35 000 ×g for 60 min at 4˚C. The pellet was rehomogenized in ice-cold 50 mM Tris buffer with Complete EDTA-free Protease Inhibitor Cocktail (pH = 7.4) followed by second centrifugation at 35 000 ×g for 60 min at 4˚C. The final membrane pellet was rehomogenized in ice-cold 50 mM Tris binding buffer with pH = 8.0 (final volume calculated to correspond to 5 million cells/ml), aliquoted (0.5 ml aliquots), and stored at -90˚C. Protein concentrations were measured by the Lowry method with some modifications [20], using bovine serum albumin as the standard.

## [³H](+)-pentazocine binding assay

The binding experiments were carried out in 96-well plate format in a final volume of 200 μL as described earlier [21] with some modifications. In all assays, 20 μL membrane suspension was added to wells containing 140 μL binding buffer and 20 μL of competitive ligand (different concentrations of PRE-084 or 10 μM haloperidol (nonspecific binding) or binding buffer (control)) and incubated for 10 min at room temperature. Then, 20 μL [³H](+)-pentazocine was added (to achieve a final concentration of 2 nM for competitive binding experiments and 0.28 to 36 nM for saturation binding curve), and samples were incubated for 180 min at 37˚C with continuous shaking. The bound radioligands were collected by rapid vacuum filtration through Millipore GF/C filters (Merck Millipore, Billerica, USA) presoaked in 0.3% polyethyleneimine. The filters were washed three times with 0.25 ml of 10 mM Tris buffer (pH = 8.0, 4˚C), and their radioactivity was measured with a Wallac MicroBeta TriLux liquid scintillation counter (PerkinElmer, Waltham, USA).

## Super-Resolution Radial Fluctuations (SRRF) microscopy

Epifluorescence and Highly Inclined and Laminated Optical sheet (HILO, pseudo-TIR [22]) imaging was conducted using an inverted microscope built around a Till iMIC body (Till Photonics/FEI, Munich, Germany), equipped with UPLSAPO 4× (NA 0.13), 20× oil (NA 0.85), 60× water (NA 1.2) and TIRF APON 60× oil (NA 1.49) objective lenses (Olympus Corp., Tokyo, Japan). The microscope was equipped with an environmental chamber (Solent Scientific Limited, Portsmouth, UK) to maintain a temperature of 37˚C with a 5% $CO_2$ supply for live-cell imaging. The samples were excited with 488 nm, 515 nm or 638 nm PhoxX laser diodes (Omicron-Laserage, Rodgau, Germany) or 561 nm acousto-optically modulated DPSS laser Cobolt Jive (Cobolt,AB, Solna, Sweden) combined in the SOLE-6 light engine (Omicron-Laserage, Rodgau, Germany) and launched into a single-mode fiber. Laser excitation was sent to a 2D Yanus scan head, which along with a Polytrope galvanometric mirror (Till Photonics/FEI, Munich, Germany), was used to position the laser focal spot in the back focal plane of the objective lens for epifluorescence or 360-pseudo-TIR illumination. Typically, imaging was performed with an excitation intensity of $\sim$ 5 mW. Emission light in colocalization experiments was spectrally separated with a ZT488/561rpc polychromatic dichroic mirror (2.0 mm substrate, Chroma Technology, Bellows Falls, USA) and a 524/628 nm BrightLine dual-band bandpass filter (Semrock Inc., Rochester, USA). For Sig1R-YFP positive control experiments, live-cell and immunofluorescence experiments in fixed-cells, additionally filter cube with a ZT514/640rpc polychromatic dichroic mirror (2.0 mm substrate, Chroma Technology, Bellows Falls, USA) and ECFP/EYFP ET dual-band bandpass filter (59017m, Chroma Technology, Bellows Falls, USA) was used. In the emission path, an additional 2× magnification was provided by a TuCam two-camera adapter (Andor Technology, Belfast, UK). Light detection was performed using an iXon Ultra 897 EMCCD camera (Andor Technology, Belfast, UK)

with an effective pixel size of 133 nm in the case of 60× objective lenses. The exposure time was held constant at 100 ms. The calibration of the microscope system and multichannel corrections were performed with a PSFcheck slide [23].

The diffraction-limited live-cell imaging experiments were performed at 37˚C in 5% $CO_2$ atmosphere using 515 nm laser for excitation. Once a region of interest (ROI) of SK-OV-3 cells with Sig1R-YFP was selected the imaging protocol in Live Acquisitions software (Till Photonics/FEI, Munich, Germany) was started, which included autofocusing steps and epifluorescence Z-stacks (20 frames, with 200 nm piezo-focusing increment) were taken every 5 min.

For SRRF experiments detecting colocalization of Sig1R-YFP (green channel) and different markers (red channel), fixed cells were used. Before the cells in the 8-well imaging chambers were fixed by using cold fixation medium containing 2% paraformaldehyde (PFA) and 0.2% glutaraldehyde (GA) in PBS (15 min at room temperature), they were incubated for 5 min with freshly prepared 0.1 M sodium borohydride solution in PBS, washed twice with PBS, and then kept at 4˚C with 0.05% sodium azide in PBS supplemented with ProLong™ Live Antifade Reagent (Invitrogen™). For imaging, selected cells with comparable expression levels of both fluorescent proteins were used. For the detection of the influence of Sig1R activation, the cells were additionally treated before fixation for different time intervals with 100 nM (+)-pentazocine. Multichannel frames were acquired in time-lapse mode (100 frames) under HILO illumination with an exposure time of 100 ms and sequential switching between 488 nm and 561 nm lasers. The SRRF images were reconstructed (100 frames per time point, magnification: 5, temporal analysis set-tings: TRA). The resolution of images was estimated by calculating the FRC [24] using the NanoJ-SQUIRREL plugin [25] and by reconstructing the original dataset separated into two different stacks composed of odd or even images. Colocalization analysis on multi-channel SRRF images was performed with the Statistical Object Distance Analysis (SO-DA) workflow [26]. Spots for colocalization analysis were extracted using a wavelet spot detector. Threshold (red channel—over 10% of the whole intensity dynamic range) and a SODA search block was applied. Spot coupling was presented as the ratio of colocalized red channel spots with green channel spots (within an increasing maximum search distance) and the whole number of detected red spots. Intracellular colocalization was quantified by analyzing images of 8 or more cells (per time point) in at least two independent experiments.

## Chemicals

Haloperidol (4-[4-(4-Chlorophenyl)-4-hydroxy-1-piperidyl]-1-(4-fluorophenyl)-butan-1-one) was purchased from Alfa Aesar, Karlsruhe, Germany. PRE-084 (2-(4-Morpholinethyl)-1-phenylcyclo-hexanecarboxylate hydrochloride) was purchased from Tocris Bioscience Bristol, UK. (+)-Pentazocine (2-dimethylallyl-5,9-dimethyl-2'-hydroxybenzomorphan) was a kind gift from Dr. C. Abate (University of Bari, Italy). [³H](+)-Pentazocine (specific activity 33.9 Ci/mmol) was purchased from American Radiolabeled Chemicals, St. Louis, MO, USA.

## Results

### [³H](+)-Pentazocine demonstrates high affinity binding in SK-OV-3-Sig1R-YFP cells

The Sig1R-YFP construct has been widely used as a model of Sig1R, and it is believed that it behaves similarly to endogenous, untagged Sig1R [6, 8]. Since Sig1R is an ER chaperone protein, in our study, we used SK-OV-3 cells, which demonstrate relatively small nuclei and a large ER membrane network. SK-OV-3 cells were transfected with Sig1R-YFP using the

BacMam system. A MOI = 3 for BacMam viruses was found to be optimal for these studies, as it revealed Sig1R-YFP expression in SK-OV-3 cells at a good level for visualization (Fig 1B). Although naive SK-OV-3 cells express endogenous Sig1R, the binding studies showed negligible specific binding of [$^3$H](+)-pentazocine (Fig 1C). We used a [$^3$H](+)-pentazocine binding assay to describe the functionality of the expressed Sig1R construct. The transfection of SK-OV-3 cells with Sig1R-YFP using the BacMam system caused a more than a ten-fold increase in the specific binding of [$^3$H](+)-pentazocine to these cell membranes (Fig 1C). In the case of naive cells, the nonspecific binding reached 60 ± 8% of the total binding, while in the case of the transfected cells, this value was only 12%. [$^3$H](+)-pentazocine retained its high binding affinity for membranes of SK-OV-3-Sig1R-YFP cells, and revealed a binding curve of simple one step binding model R + L $\Delta$ RL, with a $K_d$ = 24 ± 5 nM and receptor density $B_{max}$ = 11 ± 3 pmol/mg protein (Fig 1D, data S1A Table). The selective Sig1R agonist PRE-084 caused concentration-dependent inhibition of [$^3$H](+)-pentazocine binding with a $K_i$ value of 13 nM (Fig 1E, data S1B Table), which is similar to previously published data [27, 28].

## Localization of Sig1R-YFP in the cells

As the BacMam system was effective for the transfection of Sig1R-YFP into SK-OV-3 cells, we also used it to detect the localization of Sig1R within the cells. Cell exposure to a 515 nm laser and measurement of epifluorescence at 540 nm revealed that most of the fluorescence emitted by Sig1R-YFP was located in the ER region surrounding the nucleus (Fig 2). TIRF imaging demonstrated that no Sig1R-YFP could be detected in the plasma membrane, and the expression in the peripheral areas of cells was low (Fig 2). The Sig1R agonist PRE-084 at a concentration of 10 μM, which have been found to be optimal for the activation of Sig1R [29, 30], did not cause substantial changes in the localization of Sig1R-YFP within the cells (Fig 2 and S1 Fig). No other agonist-dependent changes in Sig1R-YFP localization were detected.

The resolution of conventional microscopic measurements did not allow us to detailly determine whether the activation of Sig1R-YFP with agonists alters its localization. As the changes were quite small and remained below the resolution limits of conventional microscopy, a super-resolution approach was proposed to provide additional information in this context. For precise localization of Sig1R in the cell, we used different fluorescent proteins as markers for specific cellular structures. For endosomes we used mCherry-Endo-14 (S2A Fig), for lysosomes—mCherry-lysosomes-20 (S2B Fig), for caveolae—Cav1-mRed (S2C Fig), for peroxisomes—mCherry-Peroxisomes-2 (S2D Fig), for mitochondria pmKate2-mito (S2E Fig, Fig 3A), and for vesicular ER—KDEL-mRFP (Fig 3B). Among these markers, only markers of mitochondria and the ER indicated the highest colocalization level with Sig1R-YFP (Fig 3E, S2E Fig and Fig 3F). This outcome was in good agreement with previous findings [12, 31], which demonstrated that Sig1R is located in subdomains of the ER that are in proximity to mitochondria.

## Mapping the Sig1R-YFP localization with SRRF and analyzing the (+)-pentazocine effect on whole cells

We have used the SRRF with a highly inclined and laminated optical sheet (HILO) illumination fluorescence microscope, which is a fast graphics processing unit enabled by the ImageJ plugin and is generally capable of achieving resolutions better than 150 nm [11]. Implementation of this method for visualization of Sig1R-YFP resulted in a detailed and sharp image that allowed us to detect Sig1R localization more precisely in fixed SK-OV-3 cells (Fig 3C and 3D). As shown in the Fig 3C and 3D, Sig1R-YFP was not evenly distributed over the whole ER, and, as also demonstrated previously [12, 31], several significantly brighter cluster structures (spots, puncta) in some ER regions were detected.

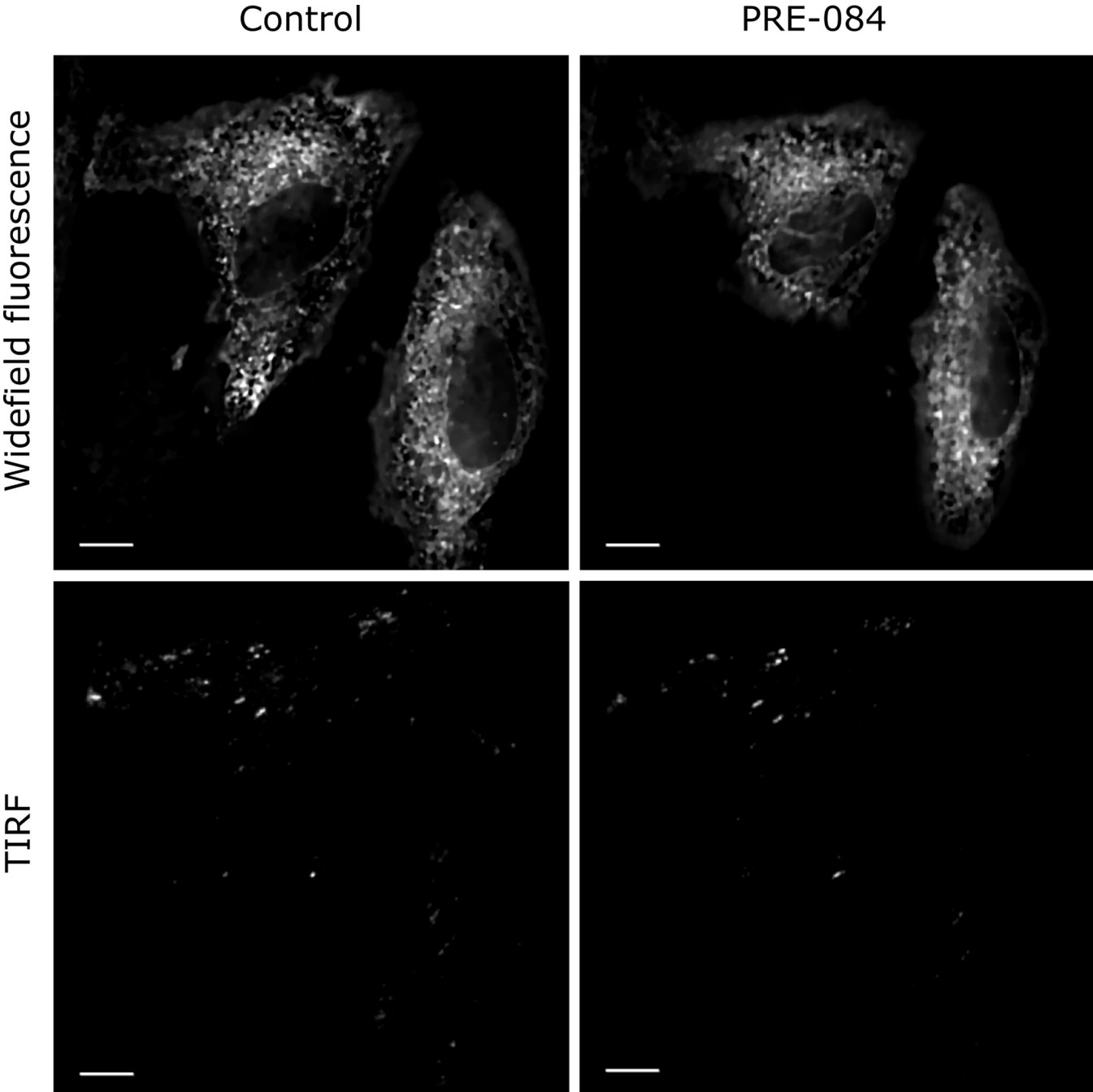

**Fig 2. Widefield fluorescence and TIRF images of Sig1R-YFP in SK-OV-3 cells after Sig1R activation with PRE-084.** SK-OV-3 cells in 8-well glass-bottom imaging chambers were transfected with recombinant BacMam baculoviruses of Sig1R-YFP (MOI = 3) 24 h before imaging. The imaging was performed on the microscope stage (heated at 37°C and supplemented with 5% $CO_2$) using 60× oil (NA 1.49) objective lenses (Olympus Corp., Tokyo, Japan) and 515 nm laser for excitation. Images were taken before (Control) and 30 min after the application of the Sig1R agonist PRE-084 (10 μm, final concentration). Scale bar = 10 μm.

SRRF enables imaging of Sig1R-YFP in live cells. However, this results in noticeably lower resolution (in our study, the minimal Fourier Ring Correlations (FRC) decreased almost two-fold) due to diffusion and movements that occur in living cells. Nevertheless, we were able to monitor the direct movement of Sig1R-YFP from the collected SRRF images (S1 Video), and

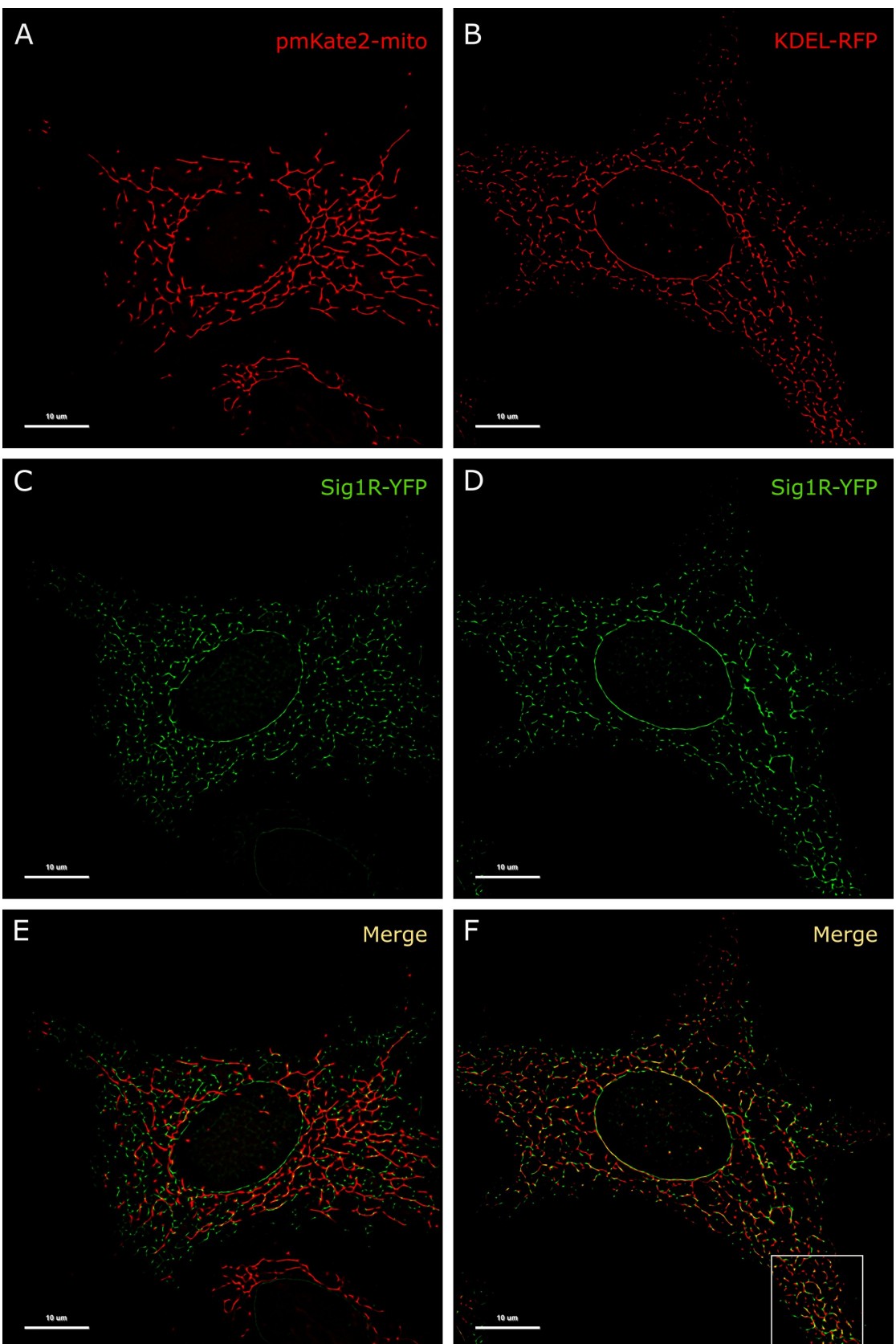

**Fig 3. Colocalization of Sig1R-YFP in SK-OV-3 cells with the markers pmKate2-mito and KDEL-mRFP.** SK-OV-3 cells expressing Sig1R-YFP and pmKate2-mito (A, C, E), or KDEL-mRFP (B, D, F) in 8-well glass-bottom imaging chambers were fixed with PFA and GA as described in the Materials and methods. Multichannel frames were acquired in time-laps mode (100 frames) under HILO illumination with an exposure time of 100 ms and sequential switching between 488 nm and 561 nm lasers with 60× oil (NA 1.49) objective lense (Olympus Corp., Tokyo, Japan). Panels E and F present SRRF images showing colocalization of Sig1R-YFP (green, C and D) with pmKate2-mito (red, A), or KDEL-mRFP (red, B), scale bar 10 μm. Square in the panel F indicates zoomed region of SRRF image (please see Fig 4A for more details).

the average velocity of the detected particles was 64 ± 21 nm/s. This is in good agreement with earlier studies, where the average velocity of the bidirectional movement of two GFP-labeled ER proteins was in the range of 70 nm/s [32].

The use of fixed cells in the experiments in which the dynamics of proteins are to be studied, required numerous prepared samples, and to obtain comparable results, all these samples of cells needed to express similar protein levels. As with conventional transient transfection, achieving equal expression is usually a challenging task; the BacMam bacu-lovirus transfection system [33] was used in addition to Sig1R-YFP for the marker proteins KDEL-mRFP and pmKate2-mito. The determination of the virus concentration for each protein enabled us to use an optimal MOI, which gave similar good expression of marker proteins in SK-OV-3 cells as it was achieved for Sig1R-YFP (Fig 1B). The use of fixed cells means that for each time point, separate measurements with separate samples must be performed. Individual samples are influenced by cells heterogeneity. However, imaging of fixed cells allowed us to avoid possible artifact associated with photobleaching and phototoxicity effects during multiple measurements or on-line monitoring, which often cause shadowing or lead to misinterpretation of the consequences of drug treatment.

By using two-color SRRF microscopy, we imaged cells coexpressing Sig1R-YFP receptors with different intracellular markers (S2 Fig). Among the studied markers, KDEL-mRFP, a marker of the ER, exhibit a pattern of spots (Fig 3B) most similar to that of Sig1R-YFP (Fig 3D). Implementation of SRRF analysis revealed that even though the high-intensity spots of these two proteins were very close to each other, they showed only partial overlap, and the majority of the "coupled" spots showed some small spatial shifts (Fig 3F, Fig 4A), which remain below the diffraction limit and could be detected with only super-resolution methods. These shifts were not unidirectional as they would be in the case of chromatic aberrations of microscopic systems. To quantify the distance between the coupled spots, we used the Statistical Object Distance Analysis (SODA) workflow [26]. Cells with a similar expression level of both fluorescent proteins were selected, and the YFP and RFP spots were localized with an automatic algorithm based on wavelet trans-formation of the image [34]. The spots at the same intensity level ($\Delta I < 10\%$ of the full intensity dynamic range) were thresholded, and SODA was used to measure spot coupling within different search distances (Fig 4B). The spot coupling of Sig1R-YFP and KDEL-mRFP achieved a level of 40% in the range of 300 nm, and it did not substantially increase with the additional increase in the search distance (Fig 4B green symbols). Considerably lower spot coupling was found between Sig1R-YFP and pmKate2-mito, which remained at 15% level in the range of 300 nm, but it continued to increase with increased search distances (Fig 4B blue filled symbols) without a clear sign of stabilization. The spot coupling of Sig1R-YFP with the other intracellular markers studied (mCherry-Endo-14 for endosomes (S2A Fig), mCherry-lysosomes-20 for lysosomes (S2B Fig), Cav1-mRed for caveolae (S2C Fig) and mCherry-Peroxisomes-2 for peroxisomes (S2D Fig) remained below 10% within the 300 nm distance and could not be reliably detected. The Pearson's correlation coefficients of these markers with Sig1R-YFP were for endosomes 0.14, lysosomes 0.15, caveolae 0.05, peroxisomes 0.03 and mitochondria 0.05, while for ER in the same super-resolution mode 0.26 (Fig 3F), but in diffraction-limited mode 0.79 (S4F Fig).

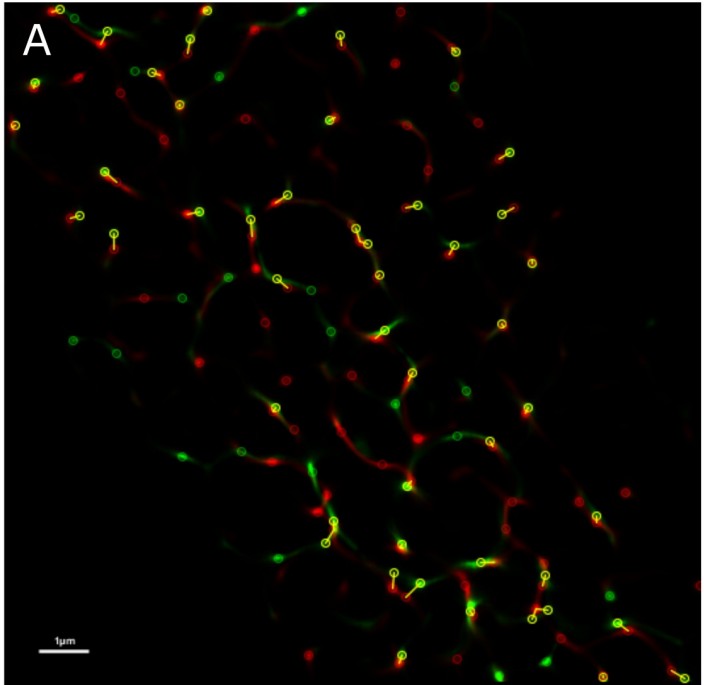

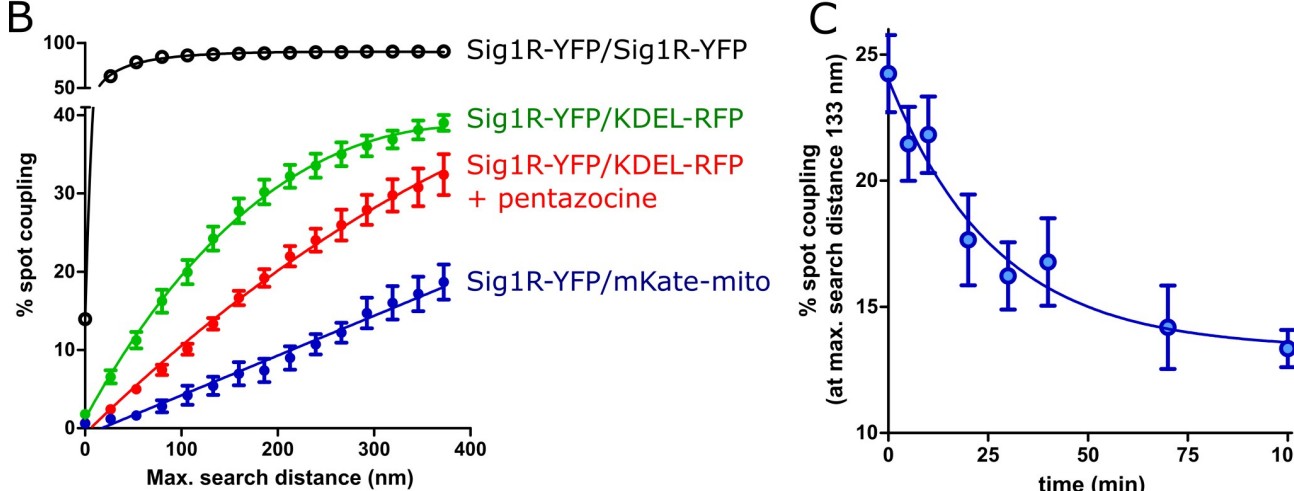

**Fig 4. The spot coupling analysis.** (A) Zoomed SRRF image of Fig 3F indicating spot coupling of Sig1R-YFP and KDEL-mRFP, which were extracted using a wavelet spot detector and marked with circles (scale bar 1 μm). (B) Dependence of the level of spot coupling of Sig1R-YFP with KDEL-mRFP (green circles), KDEL-mRFP in the presence of 100 nM (+)-pentazocine (red circles), pmKate2-mito (blue filled circles), and Sig1R-YFP (positive control, black circles) on the search distance from the receptor. The spot coupling analysis was performed with SODA workflow [26], and were presented as the ratio of red channel spots that colocalize with green channel spots to the total number of detected red spots. (C) Change in the spot coupling of Sig1R-YFP with KDEL-mRFP over time after activation of the receptor with 100 nM (+)-pentazocine. The first data point on the x-axis (0 min) indicates the baseline measurement. Spot coupling was calculated at a search distance of 133 nm, which corresponded to 5 SRRF pixels. The data were taken from images with at least 8 or more cells from at least two independent experiments.

The relatively high colocalization of Sig1R-YFP with KDEL-mRFP enabled us to study the behavior of Sig1R upon its activation with an agonist. Treatment of cells with 100 nM (+)-pentazocine caused an apparent decrease in the spot coupling of the receptor with the marker (Fig 4B red symbols). The shift was evident at all search distances studied, and in the case of activated receptors, no clear saturating threshold was achieved even at a search distance of 400

nm. Although the effects detected were not very large, the time-dependent changes in colocalization could be detected using sufficient data from the numerous samples studied in different experiments. In these experiments, the cells were fixed for each time point, and images were taken and analyzed. On average, 8 frames were taken of every well, and these usually contained 1–2 cells per frame, having on average 1000 spot localizations per cell. Approximately 300 of these spots were over the intensity threshold, which means that the spot coupling of more than 3000 objects was analyzed for every experimental point. For all images used in the analysis, the minimal FRC, which is the resolution criterion for super-resolution microscopy, was in a range between 52 and 59 nm. As we previously observed, the treatment of cells with 100 nM (+)-pentazocine caused a decrease in the level of spot coupling at all distances studied (Fig 4B). For the kinetic analysis, we selected a distance of 5 SRRF pixels (133 nm, which corresponded to 1 pixel on images without super-resolution improvement). The spot coupling level without drug treatment within this search distance was approximately 25%, and this dropped more than 10% within 100 min after the addition of the drug (Fig 4C). This decrease was time-dependent and could be described by the single exponential decay equation, revealing the first-order rate constant $(6.2 \pm 2.6) \times 10^{-4} \, s^{-1}$, corresponding to a half-life of 18.7 min (Fig 4C). To avoid artifacts and for validation of the assay, we also performed positive control experiments, where the spots of Sig1R-YFP in the same cells were re-imaged 1 min after the first image was taken, using a different set of filters and another laser. In this case, almost full coupling of Sig1R-YFP and Sig1R-YFP spots was achieved within a distance of 80 nm (Fig 4B open symbols).

## Discussion

The existence of Sig1R was first proposed in the 1970s – 1980s [35, 36]. However, the molecular mechanisms of this receptor have not been fully elucidated [37]. Sig1R was cloned in 1996 [38], while the crystal structure of full-length human Sig1R was published in 2016 [39]. Very recently, it was shown that Sig1R is an ER-localized type II membrane protein [40]. Nevertheless, there is no consensus about the dynamics of Sig1R within the cell. On the other hand, there is quite a good consensus that Sig1R is an ER protein that acts as a molecular chaperone and forms cholesterol-enriched microdomains in the ER membrane [12, 31, 40]. However, most of the evidence regarding the dynamics of Sig1R obtained with different cell imaging techniques is indirect as dynamic changes occurring remain below the diffraction limit of light microscopy and therefore cannot be clearly characterized. This study established a method to detect and analyze the intracellular dynamics of Sig1R in live cells using super-resolution imaging microscopy, which allows for shifting the detection limit below the diffraction limit. As there are no suitable antibodies or fluorescent ligands available that would allow us to monitor the dynamics of Sig1R, we used receptors tagged with YFP and transfected SK-OV-3 cells using BacMam technology. Here we must consider that YFP is a relatively large tag for Sig1R, which may influence the behavior of the receptor within the cell. However, we showed with the radioligand binding assay that the ligand binding properties of Sig1R-YFP were not affected. In addition, Yano et al. demonstrated that C-terminal-tagged Sig1R constructs show a response to ligand-induced oligomerization that is similar to that of intact Sig1R constructs [41]. Therefore, the YFP tag on the C-terminus of Sig1R likely did not have a significant impact on the intracellular dynamics of the construct used in our study. To minimize the effects associated with receptor overexpression, we used BacMam transfection technology. The detailed protocol for virus production [42] and a precise virus titration method [16] enabled optimal expression of the protein for visualization (Fig 1B), but, at the same time, prevented the formation of protein aggregates into cytoplasmic inclusion bodies. At the same time, the

number of [$^3$H](+)-pentazocine binding sites did not exceed the rat liver level [43]. This outcome is important not only in the analysis of the receptors but also the marker proteins used to determine receptor localization.

To make a step further, we had to change the conventional experimental setup: we needed super-resolution to see small changes; we needed a detection method that would enable us to see rapid changes, but we also needed a stable system that minimized the natural diffusion of proteins. Combining these three counteracting tasks was not easy but was possible. There are several super-resolution microscopy methods available [9], and here, we used SRRF analysis [11], which is a super-resolution algorithm that analyzes radial and temporal fluorescence intensity fluctuations in an image sequence to generate a super-resolution image for live-cell recordings. The signal-to-background ratio was improved by using HILO sheet illumination [22]. This setup enabled monitoring the movement of Sig1R-YFP (S1 Video) but did not allow simultaneous detection of multiple probes. Therefore, sampling was performed, and fixed cells activated at different time points with agonists were compared in this study. Nevertheless, sample heterogeneity (different transfections and probes) did not interfere with detection of the change in colocalization of Sig1R-YFP with KDEL-mRFP after treatment with the Sig1R agonist. Although the microscopy images suggested colocalization of Sig1R-YFP and pmKate2-mito; the more detailed analysis revealed that these proteins are located in proximity to each other. A recent study demonstrated similar observation that Sig1R-GFP in HEK293 cells stained with the mitochondrial marker TOM20 is in close opposition to mitochondria [12]. The spot coupling observed between Sig1R-YFP and pmKate2-mito was less than 5% at a distance of 100 nm, while it increased linearly with increasing distance (Fig 3H). In the case of KDEL-mRFP, the spot coupling at 100 nm was almost 20% and had the shape of a hyperbole with a maximum close to 60%, one half of which was achieved at a distance of 200 nm. Activation of Sig1R-YFP with agonist caused a substantial decrease in the coupling at all distances studied (Fig 4B red and green circles). The fact that we could monitor this process in time is of particular importance (Fig 4C). Although this is not the first report where the effect of activation of Sig1R has been monitored, this study demonstrates a model system that can be used to study live cell dynamics of Sig1R at a very high resolution. We acknowledge that it is merely the first step in characterizing Sig1R biochemistry and pharmacology dynamics. To date, Sig1R activity has been studied by using several high-affinity compounds and measuring dissociation from binding immunoglobulin protein (BiP also known as GRP78 or HSPA5) or evaluating the oligomerization state of Sig1R [28, 31]. However, there is no live-cell imaging assay available that could demonstrate the functional activity of the receptor or allow to study dynamic protein-protein interactions at high resolution. Of course, the assay can be modified to achieve conditions even more close to the natural and to achieve even higher resolution. The use of HaloTag [44], SNAP-Tags [45, 46] or other genetically encoded tags [47] instead of fluorescent proteins would open a wider possibilities to select fluorophores for colocalization studies and protein tracking. On the other side, the new generation fluorescent proteins like mEos4b would open possibilities to use single particle-tracking localization microscopy (sptPALM) [48] for more detailed translocation analyzes. The implementation of these and other maybe even more sophisticated methods remain for the further studies of Sig1R.

KDEL is a target peptide sequence in the C-terminal end of the amino acid structure of a protein that prevents a protein from being secreted from the ER [49]. Therefore, the peptide KDEL coupled to a fluorescent protein has been widely used as a marker of this organelle. It has been previously shown that GFP-KDEL is localized in both the vesicular sub-compartment and reticular structures of ER, while RFP-conjugated KDEL labels the vesicular sub-compartment of the ER [50]. The vesicular sub-compartment of the ER, also known as the mobile form of the ER, ER-derived punctate structure or ribosome-associated vesicles, forms direct contacts

with mitochondria [51]. Mobile ER-derived compartments are thought to be responsible for the regulation of calcium signaling and mitochondrial function [51]. Sig1R has been demonstrated to localize at the MAM [12, 31] and regulate calcium transport from the ER to mitochondria through the inositol 1,4,5-trisphosphate receptor [31]. Sig1R agonist-induced disruption of ER microdomains and a decrease in the formation of Sig1R clusters were observed in a recent study [12]. Herein, as an example, we have shown that activation of Sig1R decreases its coupling with KDEL-mRFP, which can be measured as a time-dependent process. Conventional pharmacological studies require measurements of concentration dependences of numerous ligands, agonists, partial agonists, and antagonists to provide conclusions about the protein as a drug target. Here we present the proof of principle assay system, which has a real potential to study Sig1R intracellular dynamics and automatization of the imaging for increased throughput screening [52] will allow to evaluate the activity of novel compounds acting at Sig1R.

Sig1R has been reported to be an important drug target, but most of the numerous drugs and drug candidates, that bind Sig1R with moderate affinity also have different primary targets [53]. Among these candidates are compounds with antiviral activity against hepatitis C virus (HCV) [54, 55], and COV family viruses [56], including SARS-CoV-2 [57]. Sigma receptors are among the human proteins, that have been shown to directly interact with SARS-CoV-2 proteins, particularly with the rotavirus nonstructural protein NSP6 [57]. This protein is a rotavirus protein whose function has not yet been clearly determined, but it has been proposed to be directed to mitochondria [58]. At the same time, the regulators of Sig1R are among these few pharmacological agents displaying antiviral activity in SARS-CoV-2 [57]. Therefore, it has been proposed that efficient Sig1R modulators may have potent anti-SARS-CoV-2 activity [59]. Very recently, a clinical trial demonstrated that treatment with fluvoxamine reduced the need for hospitalization of patients with early diagnosed COVID-19 [60]. However, very limited number of methods are available to study Sig1R functionality. Further development and implementation of the approach described herein for the characterization of Sig1R dynamics and pharmacological studies would lead to the acquisition of novel perspectives on this particular drug target, which may have an essential impact on SARS-CoV-2 treatment.

In conclusion, this first quantitative description of Sig1R dynamics in cells suggests new perspectives into studies of this receptor. The use of super-resolution microscopy revealed that receptor activation initiates its translocation, but the direct target/acceptor of this movement needs to be found. Further development of the assay will lead to new possibilities for the evaluation of Sig1R-targeted drugs.

## Supporting information

**S1 Fig. Activity of PRE-084 on the localization of Sig1R-YFP in SK-OV-3 cells.** SK-OV-3 cells in 8-well glass-bottom imaging chambers were transfected with recombinant BacMam baculoviruses of Sig1R-YFP (MOI = 3) 24h before the imaging. The imaging was performed on the microscope stage heated at 37˚C using 60× water (NA 1.2) objective lenses (Olympus Corp., Tokyo, Japan) and 515 nm laser for excitation. The images were taken before and 15 min and 30 min after the application of a Sig1R specific agonist PRE-084 (10 μM, final concentration) are presented as panels A, B, and C, respectively. Scale bar 20 μm.
(TIF)

**S2 Fig. Representative zoomed SRRF images of the expression of Sig1R-YFP together with intracellular markers.** Sig1R-YFP (green) together with markers of endosomes (mCherry-Endo-14, red, (A)), lysosomes (mCherry-lysosomes-20, red, (B)), caveolae (Cav1-mRed, red, (C)), peroxisomes (mCherry-Peroxisomes-2, red (D)) and mitochondria (pmKate2-mito, red,

(E)) in SK-OV-3 cells. Cells in 8-well glass-bottom imaging chambers were fixed with PFA and GA as described in the Materials and methods. Imaging of the expression of Sig1R-YFP and peroxisomes was performed in live-cells. Multichannel frames were acquired in time-laps mode (100 frames), under HILO illumination with an exposure time of 100 ms and sequential switching between 488 nm and 561 nm lasers with 60× oil (NA 1.49) objective lenses (Olympus Corp., Tokyo, Japan). Scale bar = 1 μm. The Pearson's correlation coefficients with Sig1R-YFP were for endosomes 0.14 (A), lysosomes 0.15 (B), caveolae 0.05 (C), peroxisomes 0.03 (D) and mitochondria 0.05 (E), while for ER in the same super-resolution mode 0.26 (Fig 3F), but in diffraction-limited mode 0.79 (S4F Fig).
(TIF)

**S3 Fig. Change of colocalization of Sig1R-YFP and KDEL-mRFP caused by (+)-pentazo-cine.** Colocalization demonstrated as spot coupling between Sig1R-YFP (green) and KDEL-mRFP (red) at baseline (Control, left image) and 100 min after activation of Sig1R with 100 nM (+)-pentazocine (right image). Zoomed SRRF images are presented. Multichannel frames were acquired in time-laps mode (100 frames), under HILO illumination with an exposure time of 100 ms and sequential switching between 488 nm and 561 nm lasers with 60× oil (NA 1.49) objective lenses (Olympus Corp., Tokyo, Japan). Scale bar = 1 μm. A time-dependent decrease in spot coupling was observed between Sig1R-YFP and KDEL-mRFP after activation of Sig1R with 100 nM (+)-pentazocine (middle image). The first data point on the x axis (0 min) indicates baseline measurement (Control). Spot coupling was calculated at a search distance of 133 nm, which corresponds to 5 SRRF pixels. Data are from images of at least 8 more cells from at least two independent experiments.
(TIF)

**S4 Fig. Colocalization of Sig1R-YFP with immunostained ER and mitochondria on fixed and permeabilized SK-OV-3 cells.** Cells in 8-well glass-bottom imaging chambers were fixed with glyoxal and permeabilized with Triton X-100 as described in the Materials and methods. Sig1R-YFP were expressed with BacMam (green), and organelles were detected with markers of ER (anti-PDI primary antibody) (red in A, C, D, F) or mitochondria (anti-Tom20 primary antibody) (red in G, I, J, L) and visualized with secondary antibodies labeled with Abberior Star RED. Z-stacks were acquired (100 frames) under epi-illumination with an exposure time of 50 ms and sequential switching between 515 nm and 638 nm lasers with 60× oil (NA 1.49) objective lenses (Olympus Corp., Tokyo, Japan). Stacks were deconvolved with EpiDEMIC plugin [61]. Scale bar = 10 μm for images in (A, B, C, G, H, I) and 1 μm for images in (D, E, F, J, K, L).
(PNG)

**S1 Table.** (A) Specific binding of [$^3$H](+)-pentazocine to membranes of SK-OV-3-Sig1R-YFP cells. (B) Displacement of [$^3$H](+)-pentazocine binding to membranes of SK-OV-3-Sig1R-YFP cells by PRE-084.
(PDF)

**S1 Video. Movement of the Sig1R-YFP in SK-OV-3 cells.** SK-OV-3 cells were seeded on 8-well glass-bottom imaging chambers (Zell Kontakt, Germany) at a density 20 000–25 000 cells per well 24 h before the imaging. Then the cells were transfected with Bac-Mam baculo-viruses of Sig1R-YFP (MOI = 3) and 2 h before the imaging the culture medium was exchanged by photostable cell culture medium LiveLight™ MEMO. The imaging experiments were performed at 37˚C in 5% $CO_2$ atmosphere. After selection of ROI and adjustment of HILO illumination angle the videos were acquired using continuous illumination with 515 nm laser and 10 ms exposure time per frame. SRRF images were reconstructed using NanoJ-SRRF

plugin (100 frames per time point [62], magnification: 5, temporal analysis settings: TRPPM). Bright fluorescent spots were automatically extracted using wavelet spot detector in Icy software [63] and tracked with spot tracking block which relies on Multiple Hypothesis Tracking algorithm [64] (motion mode: directed, optimization: LPsolver, filter out trajectories < 5 frames duration).
(MP4)

## Acknowledgments

We thank Dr. Peter McCormick for the pcDNA3.1(+)-Sig1R-YFP plasmid and Dr. Erik L. Snapp for providing KDEL-mRFP construct.

## Author Contributions

**Conceptualization:** Sergei Kopanchuk, Liga Zvejniece, Maija Dambrova, Ago Rinken.

**Data curation:** Sergei Kopanchuk.

**Funding acquisition:** Santa Veiksina.

**Investigation:** Sergei Kopanchuk, Edijs Vavers, Santa Veiksina, Kadri Ligi.

**Methodology:** Sergei Kopanchuk, Edijs Vavers.

**Project administration:** Maija Dambrova.

**Resources:** Santa Veiksina.

**Supervision:** Liga Zvejniece, Maija Dambrova, Ago Rinken.

**Visualization:** Sergei Kopanchuk.

**Writing – original draft:** Edijs Vavers, Ago Rinken.

**Writing – review & editing:** Sergei Kopanchuk, Edijs Vavers, Santa Veiksina, Kadri Ligi, Liga Zvejniece, Maija Dambrova, Ago Rinken.

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
