## [Decision Letter · Decision Letter 0]

4 Mar 2022

PONE-D-21-39296Intracellular dynamics of the Sigma-1 receptor observed with super-resolution imaging microscopyPLOS ONE

Dear Dr. Rinken,

Thank you for submitting your manuscript to PLOS ONE. We have now  received the reviews of two independent experts. It took some time to find suitable reviewers and I am sorry for the long delay in contacting you. After careful consideration, we feel that it has merit but does not fully meet PLOS ONE’s publication criteria as it currently stands. Therefore, we invite you to submit a revised version of the manuscript that addresses the points raised during the review process.

 Although both reviewers and myself found the study well performed and of interest, reviewers raised particular concerns that should be addressed before acceptance. Based on their reports, I notably invite you  to show that Sig1R-YFP is indeed functional and to perform an depth co localization analysis for confirming its presence in the ER.  I also recommend to show that localisation of Sig-1R-YFP is similar/equal to that of endogenous Sig-1Rs.Please submit your revised manuscript by Apr 18 2022 11:59PM. If you will need more time than this to complete your revisions, please reply to this message or contact the journal office at plosone@plos.org. Please include the following items when submitting your revised manuscript:A rebuttal letter that responds to each point raised by the academic editor and reviewer(s). You should upload this letter as a separate file labeled 'Response to Reviewers'.A marked-up copy of your manuscript that highlights changes made to the original version. You should upload this as a separate file labeled 'Revised Manuscript with Track Changes'.An unmarked version of your revised paper without tracked changes. You should upload this as a separate file labeled 'Manuscript'.

We look forward to receiving your revised manuscript.

Kind regards,

Jean-Pierre Mothet, Ph.D

Academic Editor

PLOS ONE

Journal Requirements:

“SV- Estonian Research Council grant (PSG230)

EV- European Regional Development Fund Project No. 1.1.1.2/VIAA/2/18/376 "Sigma chaperone protein as a novel drug target"

AR, EV - COST action CA 18133 ERNEST”

Reviewers' comments:

Reviewer's Responses to Questions

**Comments to the Author**

1. Is the manuscript technically sound, and do the data support the conclusions?

Reviewer #1: Yes

Reviewer #2: Yes

2. Has the statistical analysis been performed appropriately and rigorously? 

Reviewer #1: Yes

Reviewer #2: No

3. Have the authors made all data underlying the findings in their manuscript fully available?

Reviewer #1: Yes

Reviewer #2: No

4. Is the manuscript presented in an intelligible fashion and written in standard English?

Reviewer #1: Yes

Reviewer #2: Yes

5. Review Comments to the Author

Reviewer #1: Sergei Kopanchuk et al examined intracellular dynamics of the sigma-1 receptor (Sig-1R) by using the super-resolution imaging microscopy.

The Sig-1R has been proposed as a novel therapeutic target of several human diseases such as neurodegenerative disorders, drug abuse, and virus infection. The unique cell biological properties include: (1) Sig-1Rs form clusters at endoplasmic reticulum (ER) subdomains and (2) translocate upon activation by Sig-1R agonists. However, detailed characterization of subcellular localization and dynamics of Sig-1Rs wait further investigations. The technical barriers against the characterization include (1) the transfection technique which is essential for live-cell imaging may alter properties or location of the protein, especially when expressed at the ER membrane; (2) only a small portion of Sig-1Rs translocate upon their activation; (3) since Sig-1Rs move mainly on the ER membrane with a limited distance (not organelle-to-organelle translocation), high resolution is required (higher than that of conventional confocal microscopy).

Kopanchuk et al overcome a few above-mentioned barriers. They used SRRF to detect and analyze the intracellular dynamics of Sig1R. They also employed BacMam technology which was optimized to avoid the generation of protein aggregates and achieve equal expression in the majority of cells

SRRF with SODA experiments not only confirms the previously suggested Sig-1R localization (co-localization with both ER and mitochondoria markers), but also provide convincing data; namely, even though the high-intensity spots of these proteins are very close to each other, they only partially overlap, and locate within the distance of 100 nm. More importantly SRRF enables the quantitative measurement of Sig-1R agonist-induced translocation and average velocity.

The manuscript was well written and provided data should be important for a wide range of scientists (i.e., in the fields of cell biology, molecular biology and pharmacology). The experimental design, used technologies, data analyses, and data interpretation are all reasonable and scientifically sound.

Having said this, I would like to provide a few suggestions to increase the quality of the paper.

1. Figure 3 includes the most important data, but its resolution is not high enough (This problem might be caused by its conversion to pdf during the processing of the submission). Please increase the quality of the image.

2. It may be important to provide data that suggests the subcellular Sig-1R-YFP is similar/equal to that of endogenous Sig-1Rs. The subcellular membrane fractionation followed by western blotting should answer to this question.

Reviewer #2: Major Comments

1. The previous authors who verified that Sig1R tagging with YFP did not render it non-functional tested it in NG108-15 cells (Hayashi & Su, J Pharmacol Exp Ther, 2003) and MEF cells (Wong et al., Mol Pharmacol. 2016). Can the authors provide evidence that this same tagging does not alter function in SK-OV-3 cells?

2. As shown in figure1B, there is little specific binding for endogenous Sig1R by pentazocine. Does this mean the endogenous Sig1R is non-functional? If it is functional, why does the Sig1R-YFP binding show an order of magnitude about 9 times that of the endogenous?

3. The authors should re-work Figure2. If these are fixed cells, then it needs to show at least DAPI stained nuclei as well as ER, mitochondria or Golgi stained cells. Provide the merged imaged for these different stains in addition to the Sig1R-YFP fluorescent cells.

The current data doesn’t prove that Sig1R-YFP is localised to the ER. It could be simply cytoplasmic localised without preference of the ER. A colocalization analysis is the minimum needed to verify its association with other cellular organelles in this data.

4. The authors should provide a Pearson’s correlation co-localisation analysis for Fig S2. This is absolutely essential!

5. The authors should consider the merits of doing single molecule imaging with Sig1R tagged to wither Halo or mEos. This should be mentioned in the discussion segment.

Minor Comments

1. The authors should provide a more detailed data for the expression of Sig1R-YFP than what is shown in Fig.1A. Consider providing confocal images taken on a 63X of 100X objective with some nuclear and ER immunolabelling.

2. The authors should provide a schema of the (pentazocine) binding assay before showing Fig.1B. It is a bit jarring. Also, the authors should provide raw data of the binding assay done on the naïve cells, Sig1R-YFP expressing cells and the PRE-084 treated cells (graphs). Make it a supplementary to Fig.1C & D.

3. The authors should consider using a 100X Obj in TIRF configuration to verify absence of Sig1R-YFP. Also the data will be more convincing if the authors label the cell with Sig1 tagged with multiple tags, for example, GFP and mCherry/mRuby and show by TIRF imaging that these are also not localised to the plasma membrane.

4. Are the cells treated with PRE-084 the same as in the control images? If yes, it seems PRE-084 has effects in shrinking the cell.

5. In line 273, the authors should provide a reference where 10 µM of PRE-084 was used. If no reference is available, then the authors should provide a concentration gradient data, for example, 0 -150 uM with time showing the (absence/presence) of effect PRE-084 on Sig1-YFP in these cells.

6. The authors should re-word lines 290 – 293, such that it ends with Fig S2D) instead of Fig S2C).

7. The authors should consider placing the colour scheme naming in Fig.3H and not only place this information inside the figure legend.

6. PLOS authors have the option to publish the peer review history of their article (what does this mean?). If published, this will include your full peer review and any attached files.

Reviewer #1: **Yes: **Teruo Hayashi

Reviewer #2: No

---

## [Author Response · Author response to Decision Letter 0]

27 Apr 2022

Dear Editor, dear reviewers,

Thank you very much for the effort, kind words, and useful and insightful suggestions. We have discussed the suggestions with all the authors and formulated answers and performed changes in the manuscript, which are described in detail in the file "Response_to_reviewers_REV1" uploaded to the Editorial Manager.

---

## [Editor Report · Decision Letter 1]

3 May 2022

Intracellular dynamics of the Sigma-1 receptor observed with super-resolution imaging microscopy

PONE-D-21-39296R1

Dear Dr. Rinken,

We’re pleased to inform you that your manuscript has been judged scientifically suitable for publication and will be formally accepted for publication once it meets all outstanding technical requirements.

Kind regards,

Jean-Pierre Mothet, Ph.D

Academic Editor

PLOS ONE
---

## [Editor Report · Acceptance letter]

10 May 2022

PONE-D-21-39296R1 

Intracellular dynamics of the Sigma-1 receptor observed with super-resolution imaging microscopy 

Dear Dr. Rinken:

I'm pleased to inform you that your manuscript has been deemed suitable for publication in PLOS ONE. Congratulations! Your manuscript is now with our production department. 

Kind regards, 

on behalf of

Dr Jean-Pierre Mothet 

Academic Editor

PLOS ONE